# Is There a Correlation between Multiparametric Assessment in Ultrasound and Intrinsic Subtype of Breast Cancer?

**DOI:** 10.3390/jcm10225394

**Published:** 2021-11-19

**Authors:** Magdalena Gumowska, Joanna Mączewska, Piotr Prostko, Katarzyna Roszkowska-Purska, Katarzyna Dobruch-Sobczak

**Affiliations:** 1Department of Radiology, Maria Sklodowska-Curie National Research Institute of Oncology, 02-034 Warsaw, Poland; m.ewabielawska@gmail.com (M.G.); kdsobczak@gmail.com (K.D.-S.); 2Nuclear Medicine Department, Medical University of Warsaw, 02-097 Warsaw, Poland; 3Statistician, 00-001 Warsaw, Poland; prostko.p@gmail.com; 4Department of Pathomorphology, Maria Sklodowska-Curie National Research Institute of Oncology, 02-034 Warsaw, Poland; roszkowskapurska@gmail.com

**Keywords:** breast cancer, ultrasound features, multiparametric assessment

## Abstract

Molecular profile of breast cancer provides information about its biological activity, prognosis and treatment strategies. The purpose of our study was to investigate the correlation between ultrasound features and molecular subtypes of breast cancer. From June 2019 to December 2019, 86 patients (median age 57 years; range 32–88) with 102 breast cancer tumors were included in the study. The molecular subtypes were classified into five types: luminal A (LA), luminal B without HER2 overexpression (LB HER2−), luminal B with HER2 overexpression (LB HER2+), human epidermal growth factor receptor 2 positive (HER2+) and triple negative breast cancer (TNBC). Histopathological verification was obtained in core biopsy or/and post-surgery specimens in all cases. Univariate logistic regression analysis was performed to assess the association between the subtypes and ultrasound imaging features. Experienced radiologists assessed lesions according to the BIRADS-US lexicon. The ultrasound scans were performed with a Supersonic Aixplorer and Supersonix. Based on histopathological verification, the rates of LA, LB HER2−, LB HER2+, HER2+, and TNBC were 33, 17, 17, 16, 19, respectively. Both LB HER2+ and HER2+ subtypes presented higher incidence of calcification (OR = 3.125, *p* = 0.02, CI 0.0917–5.87) and HER2+ subtype presented a higher incidence of posterior enhancement (OR = 5.75, *p* = 0.03, CI 1.2257–32.8005), compared to other subtypes. The calcifications were less common in TNBC (OR = 0.176, *p* = 0.0041, CI 0.0469–0.5335) compared to other subtypes. There were no differences with regard to margin, shape, orientation, elasticity values and vascularity among five molecular subtypes. Our results suggest that there is a correlation between ultrasonographic features assessed according to BIRADS-US lexicon and BC subtypes with HER2 overexpression (both LB HER2+ and HER2+). It may be useful for identification of these aggressive subtypes of breast cancer.

## 1. Introduction

Breast cancer (BC) is characterized by marked heterogeneity, regarding clinical and radiographic presentation, as well as response to therapy. It is largely caused by polymorphism of histological types and variable molecular profile of specific BC types. Introduction of immunohistochemistry testing (IHC) to routine practice resulted in a significant progress in understanding BC biology. IHC testing is the basis for classification of four main BC subtypes: luminal A (LA), luminal B (LB), human epidermal growth factor receptor 2 positive (HER2+) and triple negative (TNBC, so called basal breast cancer). LB was further subdivided into two subtypes: LB HER2− and LB with HER2 overexpression (LB HER2+). Determination of a specific cancer subtype allows for treatment optimization (surgery or preoperative chemotherapy). On the other hand, BC subtype is one of the prognostic factors, along with tumor size, grade, lymph node involvement and Ki67 proliferation rate [1]. Early diagnosis in patients with subtypes characterized by aggressive biology (TNBC and HER2+) and implementation of preoperative chemotherapy, is of utmost importance.

Diagnostic workup of BC utilizes ultrasound scans (US), mammography (MMG) including contrast-enhanced spectral mammography and magnetic resonance imaging (MRI). The above-mentioned BC subtypes are characterized by specific features in different imaging modalities [2,3,4,5,6,7,8,9,10,11,12,13]. (Table 1).

Luminal cancers commonly appear as densities with spicular margins, while TNBC and HER2+ cancer often exhibit blurred margins in MMG scans [2]. Other authors emphasize that TNBC subtype (in particular high-grade ones, G3) appear in MMG as well delineated densities without microcalcifications, and therefore may imitate benign lesions [3]. MRI reveals stronger background parenchymal enhancement (BPE) in TNBC, weaker in luminal B (HER2−) type [4]. Significant differences between specific BC types were found in multiparametric MRI (dynamic contrast enhanced (DCE), diffusion weighted imaging (DWI) and spectroscopy)) [5]. In MRI, TNBC subtype usually appears as unifocal necrotic masses with heterogeneous marginal enhancement and increased signal intensity in T2-weighted images, which corresponds to necrosis [6]. Yang et al. revealed that microlobular margins are more commonly found in US scans of this BC subtype [7]. This was supported by subsequent studies and its low echogenicity, irregular shape, lack of calcifications, posterior enhancement and lack of spicules were emphasized [8]. On the other hand, Li et al. demonstrated that high grade (G3) TNBC type more commonly appears as lesions with irregular shape [9].

Sonoelastography scans demonstrated that tumors with HER2 overexpression exhibit higher Young’s modulus values in shear wave elastography (SWE) than LA tumors in one study [10], while other study [11] revealed similar, high Young’s modulus values in SWE for all molecular BC subtypes, with the exception of tubular BC [12].

The aim of our study was to identify correlation between US images of BC and its molecular profile utilizing various parameters of assessment of US scans, including sonoelastography and lesion vascularity pattern.

## 2. Materials and Methods

### 2.1. Patients

Patients referred for US imaging to the Second Department of Radiology of National Institute of Oncology (NIO) in Warsaw with focal lesions BIRADS 4 and BIRADS 5, qualified for biopsy, were subjected to this retrospective analysis.

US examinations and biopsies were performed from June 2019 to December 2019 by three experienced radiologists in breast US imaging (M.G., J.M. and K.D.S.—respectively 6, 10 and 20 years of experience). The study inclusion criteria were as follows: (1) focal lesions confirmed by histopathological analysis of core needle biopsy specimens as BC with immunohistochemical assessment, (2) lesions visible in B-mode US imaging with assessment of rigidity by sonoelastography and the lesion vascularity. All patients had no previous history of a breast cancer. Static images and videos with B-mode, color Doppler and sonoelastography imaging were recorded on the external disc.

The study protocol was approved by the Scientific Council of NIO and the Ethics Committee of Maria Sklodowska—Curie National Research Institute of Oncology, Poland (no 49/2018, 27 September 2018). Patients provided a written consent to take part in the study.

### 2.2. Sonoelastography

US scans were obtained using Aixplorer scanner and a linear transducer L4-18MHz with shear wave elastography (SWE) as well as SuperSonix scanner and a linear transducer L4-14MHz utilizing strain elastography (SE).

Focal lesions were evaluated in radial and antiradial planes. They were classified according to BIRADS 2013 and Polish Ultrasound Society (PUS) classifications [13,14]. The following features of focal lesions were assessed: shape, orientation, margins, echogenicity, an effect behind the lesion, presence of edema, presence of vessels and stiffness (Table 2).

Sonoelastography was performed according to EFSUMB (European Federation of Societies for Ultrasound in Medicine and Biology) [15] and ACR BIRADS (American College of Radiology, Breast Imaging-Reporting and Data System) guidelines [13]. SE utilized five points Tsukuba scale (1 = strain in the entire lesion, 2 = strain is seen within most of the area, 3 = strain appears only in the periphery, 4 = no strain within the lesion, 5 = no strain is measured within the lesion nor in the surrounding tissues) and SWE analyzed maximum Young’s modulus value (E max) in the lesion, including up to 2 mm of the tissue around tumor margin, ROI diameter 2 mm.

In ACR (American College of Radiology) elasticity assessment is divided in three categories, without detailed indication. According the EFSUMB recommendation and WFUMB guidelines [15,16], we scheduled in the study following division:Soft (incorporate: Tsukuba1,Tsukuba2, E max < 80 kPa)Intermediate (incorporate: Tsukuba3, E max > 80 kPa < 160 kPa)Hard (incorporate: Tsukuba 4,Tsukuba5, E max > 160 kPa)

Figure 1 shows examples of sonoelastography images.

### 2.3. Histopathological Verification

All patients underwent US-guided core needle biopsy of focal lesions having provided consent to the procedure. Adjacent tissues were anesthetized with 4 mL of 2% lignocaine; the skin was incised with a scalpel and samples were collected from the tumor (from 3 to 5). The tissue material was formalin fixed and paraffin embedded followed by staining using specific antibodies to assess the immunohistochemistry profile of the tumor. Steroid receptors (ER and PGR) were assessed using Allred scale that utilizes the rate of stained cellular nuclei [17]. IHC was also used to classify HER2 on a 4-point scale, where 0, 1 indicated no HER2 overexpression, 3-presence of HER2 overexpression. Tumors with borderline value (2+) underwent fluorescence in situ hybridization (FISH) assessment. If HER2 index on chromosome 17 was above 2.2, the tumor was classified as HER2 positive (for details see Appendix A).

Histopathological verification was performed by a specialized pathologist with 25 years of experience in assessment of breast tumors (K.R.P). Based on IHC and FISH as well as ER, PR, HER2 and Ki-67 statuses, tumors were classified into subtypes according to St. Gallen guidelines (Table 3).

### 2.4. Statistical Analysis

The results presented in this paper were obtained using R statistical software, version 3.6.3 (R Core Team 2020. R: A language and environment for statistical computing. R Foundation for Statistical Computing, Vienna, Austria. URL: https://www.R-project.org/). The quantitative assessment of the relationship between the individual features of the ultrasound image and the incidence of cancer subtypes was performed using the odds ratio (OR), *p*-value and confidence interval (CI) obtained from a one-dimensional logistic regression model with a logit link function. In the same model, a given feature of US was adopted as the explanatory variable, and the role of the dependent variable was played by a binary variable of the form: 1 when LA cancer, 0 when a different cancer subtype; 1 when the cancer is HER2+, 0 when a different cancer subtype, etc., or a binary variable corresponding to a grouping of several subtypes, e.g., 1 when LB HER2+, HER 2+ and TNBC, 0 when LA and LB HER2−. The statistical significance of the results obtained in this way was interpreted assuming a type I error of 5%. *p*-values and confidence intervals do not take into account the correction for multiple testing.

## 3. Results

The study group included 102 tumors BIRADS-US 4 and BIRADS-US 5. The study enrolled 86 patients aged 32 to 88 years, in whom histopathological investigation identified cancer. Median age in the LA group was 65.0 years (interquartile range [IQR] 50.5–77.5), LB HER2− 52.5 years (IQR 38.0–65.0), LB HER2+ 57.0 years (IQR 46.25–68.5), HER2+ 56.0 years (IQR 43.0–65.0) and TNBC 56.5 years (IQR 46.25–67.5), Figure 2.

Horizontal lines represent median (black), the bottom and the top of the boxes represent the upper and the lower quartile, the whiskers represent 10th and 90th percentile, while the solid circles represent the individual data points.

US imaging assessment of 102 breast tumors, including: 33 LA cancers, 17 LB HER2− cancers, 17 LB HER2+ cancers, 16 HER2+ and 19 TNBC cancers. Figure 3 presents a percentage of individual cancer subtypes. LA subtype dominated in the study group.

Tumors were assessed using BIRADS-US lexicon, according to ACR BIRADS Atlas 2013 [13]. Table 4 presents detailed results of the analysis. Of note, irregular shape, parallel orientation, mixed and/or reduced echogenicity (hypoechogenic) dominated in the study group. Only two lesions (HER2+) were characterized by well delineated margins, while other lesions exhibited blurred margins.

Statistical analysis of risk of cancer development depending on age revealed that for each year of age the risk of LA cancer increased by 4.6% (versus all other cancer subtypes combined). However, when age rose by 10 years, the risk of LA cancer increased by approx. 57%. Other cancer subtypes did not exhibit age dependence (Table 5).

Considering biology of specific IHC subtypes of BC, they were divided into two groups:Aggressive subtypes, including: LB HER2+, HER2+ and TNBCLuminal, including subtypes: LA and LB HER2−

We found statistically significant association between the tumor size and the two groups of BC subtypes, i.e., aggressive and luminal (*p* = 0.0252, OR 1.045, CI 1.0078–1.0896). As shown in Figure 4, larger tumors tend to be more aggressive. In particular, for a 1 mm increase in size, the risk for aggressive type increases by 4.5%. Furthermore, when tumor dimension increased by 10 mm, the risk for aggressive type increased by 55%.

Horizontal lines represent median (black), the bottom and the top of the boxes represent the upper and the lower quartile, the whiskers represent 10th and 90th percentile, while the solid circles represent the individual data points.

The analysis assessing correlation between morphological features and cancer type showed that posterior enhancement was more common in HER2+ subtypes (including HER2+ and LB HER2+) (Table 6). The odds ratio for HER2+ subtype (versus other subtypes) is higher (OR = 5.75) when there is posterior enhancement vs. no effect behind the lesion (Table 6).

Differences were also found when cancers were classified as luminal or aggressive. The OR for an aggressive cancer is 29.25-fold higher when there is a posterior enhancement vs. no effect behind the lesion (Table 7).

Figure 5 shows typical features of HER2+ subtype.

Table 9 presents results for TNBC subtype, where calcifications were found less commonly. The OR for TNBC (vs. other subtypes) were 5.68-fold lower when calcifications were found.

Figure 6 shows an example of TNBC.

No statistically significant differences were found for other features of the BIRADS lexicon for specific tumors, in particular lesion margins, shape, orientation, stiffness in sonoelastography or echogenicity (see Appendix A).

## 4. Discussion

Analysis of IHC subtypes of breast cancer is currently used in routine clinical practice. Specific cancer subtypes differ not only in terms of microscopic features, but also exhibit differences in imaging studies.

The most common LA cancer has the best prognostic indices among all five BC subtypes [18]. In our study group, this subtype was also the most common and occurred in the oldest patients, while mean age of patients with other BC subtypes was lower. We did not find any typical features that would differentiate LA BC from other subtypes. However, Zhang L et al. demonstrated that hyperechoic halo around the tumor and acoustic shadowing or lack of acoustic posterior effect were more common in this subtype [12]. Hyperechoic halo accompanying BC may represent edema around the tumor (of inflammatory or lymphatic origin) or poorly delineated spicules. Spicules visibility strongly depends on echogenicity of adjacent tissues—the surrounding hyperechoic fibrotic tissue makes spicules hypoechoic, while surrounding adipose tissue (as a reference area) may make them hyperechoic. The updated ACR BI-RADS recommendations do not include a halo around a tumor as a feature [13].

TNBC poses a special challenge for radiologists and oncologists. It is an aggressive cancer with high cellular proliferation rate, common *TP53* mutations and adverse clinical prognosis [19]. Its treatment includes neoadjuvant chemotherapy [20,21] that results in high pathologic complete response (pCR) rate. The risk of disease recurrence within 3–4 years in patients with residual disease in breast or axillary lymph nodes is high. It is a heterogeneous group of tumours, characterized by variable gene expression pattern (6 subtypes of this BC were identified), resulting in variable response to therapy [22]. This also concerns US images. Zhang L. et al. described this subtype using two separate patterns of US images. One was characterized by irregular shape, lobular margins, lack of calcifications and vessels, in second pattern lesions characteristics were oval shape, microlobular margins and lack of visible vessels [12]. The latter US pattern may imitate potentially benign BIRADS 3 lesions (e.g., fibroadenomas).

In our study, lack of calcifications differentiated TNBC from other BC subtypes. Other authors also reported similar results—Dogan et al. found calcifications only in 4.5% of TNBC cases [23]. Circumscribed margins are another feature emphasized in the literature—Dogan et al. found them in 32% of lesions, while Ko et al. showed them in 57% of TNBC cases in their study [24].

Posterior enhancement and calcifications were other features that differentiated BC subtypes. Our study confirmed that calcifications and posterior enhancement were more common in HER2+ subtypes (LB HER2+ and HER2+). Nowadays these subtypes are being treated with preoperative chemotherapy. They are characterized by increased cellular proliferation rate and neovascularization as well as good response to therapy targeting HER2. Posterior enhancement was also shown by authors to be more common in aggressive cancers (LB HER2+, HER2+ and TNBC) and tumors with posterior acoustic enhancement were found to be more cellular and tend to be high-grade [9,19]. Posterior acoustic enhancement is a finding that can be associated with a variety of entities, including normal anatomic structures, simple cysts, complicated cysts, fibroadenoma, nodular sclerosing adenosis, papilloma, complex cystic mass, invasive ductal carcinoma, and lymphoma [25].

We did not find any differences in margin appearance for specific BC subtypes, including TNBC; circumscribed margins were found only in 2/102 cancer lesions. Both were HER2+ BC with high Ki67 proliferation rate (30% and 80%). A metaanalysis including 12 studies with a total of 2741 HER2+ BC tumors revealed that presence of microcalcifications is a feature that increases OR for this subtype (OR 2.45, CI 0.23–0.69), and circumscribed margins reduced OR for this subtype (OR 0.66, CI 0.43–1.02) [26].

Breast cancer microenvironment, assessed indirectly in US scans using its stiffness (SWE imaging), is another important parameter discussed in literature. It can differentiate specific BC subtypes. Yoo et al. showed that increased (mean) tumor stiffness in SWE imaging is correlated with hypoxia of the tumor and surrounding tissues (assessed using expression of an endogenous GLUT1 protein) and is an independent prognostic biomarker. Increased stiffness is associated with adverse prognosis [27]. The highest stiffness was found in TNBC tumors (13/82) in this study. Li Z. et al. on the other hand did not find differences in stiffness between specific BC subtypes [8]. Evans et al. also did not find differences in stiffness between invasive cancer subtypes using SWE imaging in a group of 137 tumors [11]. Tubular cancers were the exception: they exhibited lower stiffness indices (23% of tubular cancers had E max < 50 kPa vs. 6% of NST cancers). The authors of this paper did not use IHC classification, but categorized tumors according to their pathological structure (e.g., NST, lobular, mucous or other). We also did not find differences in tumor stiffness for specific BC subtypes. Only 14/102 breast tumors were classified as soft, while the rest of them were hard or of intermediate stiffness.

The limitations of our study are the following: it was based on analysis of lesions from a single center, and it was a retrospective study. Large, multicenter studies could aid in creating models predicting molecular subtype of BC with higher accuracy.

## 5. Conclusions

Results of our study indicate that presence of certain features can suggest TNBC and HER2+ subtypes: calcifications in HER2+ tumors and posterior enhancement in aggressive BC. Calcifications were less common in TNBC subtypes. This information may contribute to improved identification of these BC subtypes and affect decisions concerning indications for biopsy.

The other features of US imaging, such as: lesion margins, shape, orientation, stiffness assessed using sonoelastography or echogenicity, did not differ between specific tumor subtypes.

## Figures and Tables

**Figure 1 jcm-10-05394-f001:**
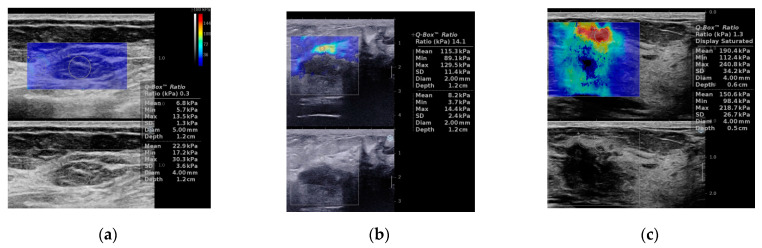
Sonoleastography. Shear wave elastography (SWE), (**a**–**c**) examples of tumors. (**a**) 65-year patient with TNBC, soft in elastography (E max 13.5 kPa). (**b**) 52-year patient with LB breast cancer, intermediate stiffness (E max 129.5 kPa). (**c**) 67-year patient with HER2+ breast cancer, stiff on elastography (E max 240 kPa). Strain elastography (SE), (**d**,**e**) examples of tumors. (**d**) 32-years patient with subtype LB HER2+ breast cancer, Tsukuba1. (**e**) Ultrasound examination in a 65-years old patient with TNBC, Tskububa5. LB HER2+ = luminal B with HER2 overexpression; HER2+ = human epidermal growth factor receptor 2 positive; TNBC = triple negative breast cancer.

**Figure 2 jcm-10-05394-f002:**
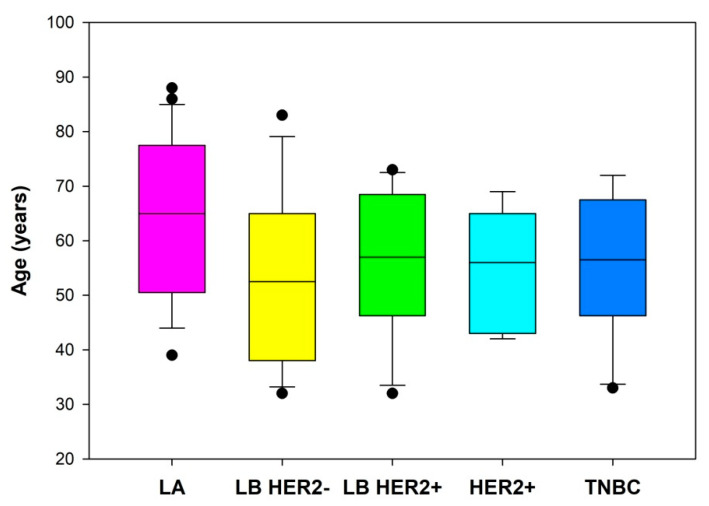
Age of patients with specific breast cancer subtypes. LA = luminal A; LB HER2− = luminal B without HER2 overexpression; LB HER2+ = luminal B with HER2 overexpression; HER2+ = human epidermal growth factor receptor 2 positive; TNBC = triple negative breast cancer.

**Figure 3 jcm-10-05394-f003:**
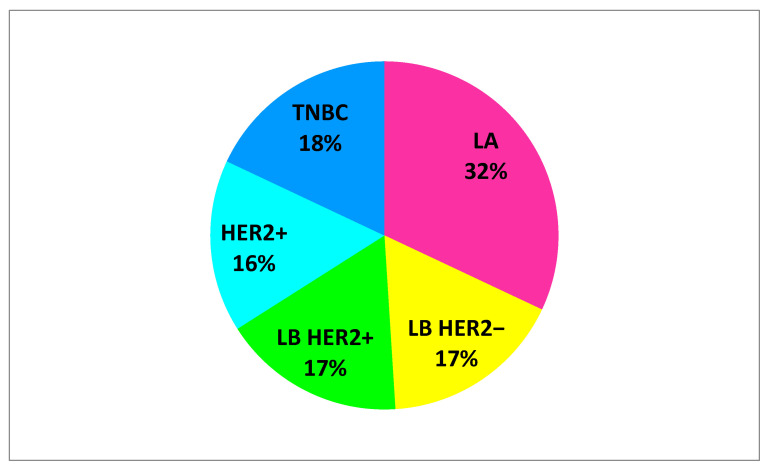
Percentage of specific breast cancer subtypes. LA = luminal A; LB HER2− = luminal B without HER2 overexpression; LB HER2+ = luminal B with HER2 overexpression; HER2+ = human epidermal growth factor receptor 2 positive; TNBC = triple negative breast cancer.

**Figure 4 jcm-10-05394-f004:**
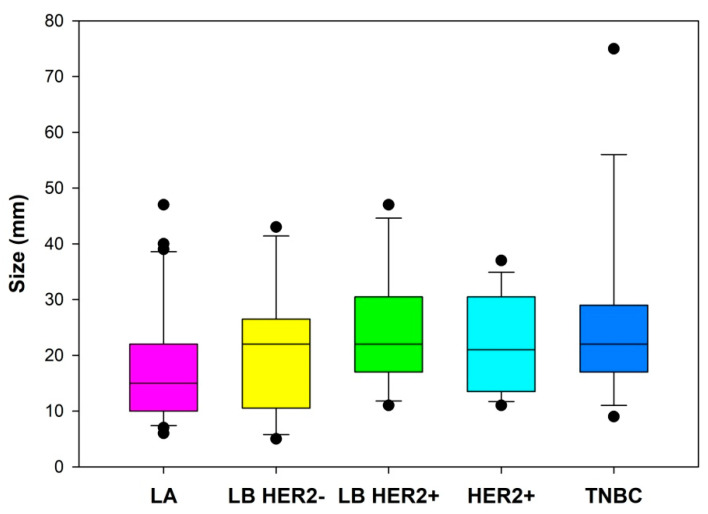
Size of tumors of specific breast cancer subtypes. LA = luminal A; LB HER2− = luminal B without HER2 overexpression; LB HER2+ = luminal B with HER2 overexpression; HER2+, = human epidermal growth factor receptor 2 positive; TNBC = triple negative breast cancer.

**Figure 5 jcm-10-05394-f005:**
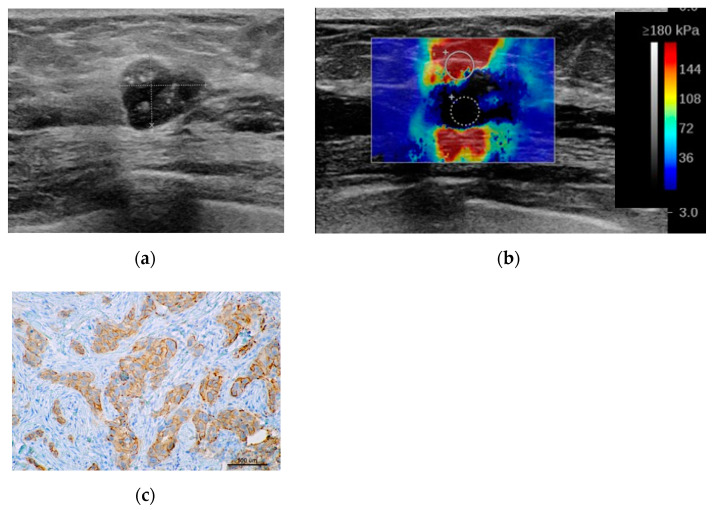
Typical HER2+ tumor. (**a**) B-mode shows posterior enhancement and calcifications. (**b**) On shear wave elastography (SWE) increased stiffness is visible, Emax 300 kPa. (**c**) IHC HER2(2+) membrane staining (magnification 10×). HER2+ = human epidermal growth factor receptor 2 positive; IHC = immunohistochemistry.

**Figure 6 jcm-10-05394-f006:**
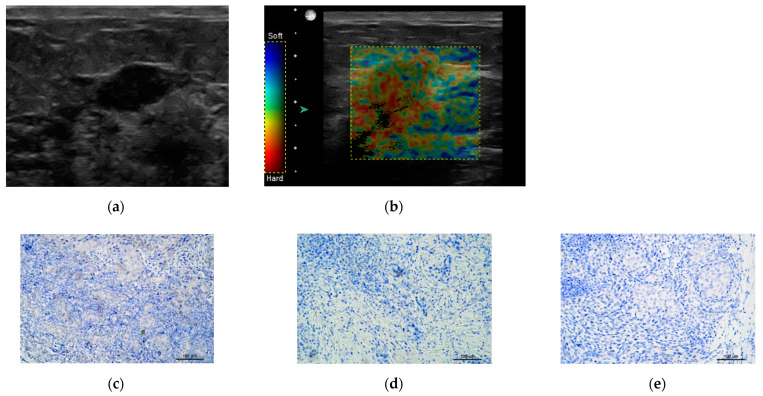
Triple negative breast cancer TNBC (**a**) B-Mode hypoechoic, without calcifications. (**b**) Sonoelastography showssoft tumor Tsukuba2. Lower panel IHC staining: (**c**) HER2 negative, (**d**) PG negative, (**e**) ER negative. IHC = immunohistochemistry, ER = estrogen receptor, PR = progesterone receptor, HER2 = human epidermal growth factor receptor.

**Table 1 jcm-10-05394-t001:** Correlation between radiologic features and BC subtypes—literature review.

MMG	Luminal cancers commonly appear as densities with spicular margins, while TNBC and HER2+ cancer often exhibit blurred margins [2] TNBC subtype (in particular high grade ones, G3) appear as well delineated densities without microcalcifications, and therefore, may imitate benign lesions [3]
US	Microlobular margins are more commonly found in TNBC [7] TNBC presented as microlobulated, markedly hypoechoic masses with an abrupt interface [8] High grade (G3) TNBC type more commonly appears as lesions with irregular shape [9] Tumors with HER2 overexpression exhibit higher Young’s modulus values in shear wave elastography (SWE) than LA tumors [10] Similar, high Young’s modulus values in SWE for all molecular BC subtypes, with the exception of tubular BC [11] Particular sets of features for individual breast cancer types [12]
MR	MRI reveals stronger background parenchymal enhancement (BPE) in TNBC, weaker in luminal B (HER2−) type [4] TNBC subtype usually appears as unifocal necrotic masses with heterogeneous marginal enhancement and increased signal intensity in T2-weighted images, which corresponds to necrosis [6]

MMG = mammography, US = ultrasound scan, MRI = magnetic resonance; LA = luminal A; LB HER2− = luminal B without HER2 overexpression; LB HER2+ = luminal B with HER2 overexpression; HER2+ = human epidermal growth factor receptor 2 positive; TNBC = triple negative breast cancer.

**Table 2 jcm-10-05394-t002:** Analyzed features of US images.

Shape	Oval
	Round
	Irregular
Orientation	Parallel
	Non-parallel
Margin	Circumscribed
	Non-circumscribed
	Indistinct
	Micro/macrolobulated
	Angular
	Spiculated
Echo pattern	Anechoic
	Hyperechoic
	Complex cystic and solid
	Hypoechoic
	Isoechoic
	Heterogeneous
Postrior features	No posterior features
	Enhancement
	Shadowing
	Combined pattern
Calcifications	Absent
	In a mass
	Outside of a mass
	Intraductal
	Hyperechogenic foci
Skin	Normal
	Thickening > 2 mm
Edema	Absent
	Present
Vascularity	Absent
	Internal
	Vessels in a rim
	Outside of a mass
Elastography	Soft < 80 kPa
	Intermediate 80–160 kPa
	Hard > 160 kPa

**Table 3 jcm-10-05394-t003:** Molecular classification of breast cancer.

LA	ER+ and PR ≥ 20% HER2− Ki-67 < 20%	
LB	LB HER2−	LB HER2+
	HER2− ER+ and any of the following: Ki-67 ≥ 20% and/or PR− or <20%	HER2+ ER+ Any Ki-67 and PR
HER2+	ER and PR- HER2+	
Basal-like (TNBC)	ER and PR− HER2−	
Special Types	ER+ (cribriform, tubular, mucous) ER− (apocrine, medullary, adenoid cystic carcinoma, metaplastic)	

ER = estrogen receptor, PR = progesterone receptor, HER2 = human epidermal growth factor receptor, LA = luminal A; LB HER2− = luminal B without HER2 overexpression; LB HER2+ = luminal B with HER2 overexpression; HER2+ = human epidermal growth factor receptor 2 positive; TNBC = triple negative breast cancer.

**Table 4 jcm-10-05394-t004:** Characteristics of specific breast cancer subtypes using BIRADS lexicon.

		LA	LB HER2−	LB HER2+	HER2+	TNBC
Shape						
Oval		0/33 (0%)	0/17 (0%)	0/17 (0%)	1/16 (6%)	0/19 (0%)
Round		2/33 (6%)	0/17 (0%)	0/17 (0%)	0/16 (0%)	0/19 (0%)
Irregular		31/33 (94%)	17/17 (100%)	17/17 (100%)	15/16 (94%)	19/19 (100%)
Orientation						
Parallel		25/33 (76%)	16/17 (94%)	16/17 (94%)	15/16 (94%)	15/19 (79%)
Non-parallel		8/33 (24%)	1/17 (6%)	1/17 (6%)	1/16 (6%)	4/19 (21%)
Margin						
Circumscribed		0/33 (0%)	0/17 (0%)	0/17 (0%)	2/16 (12.5%)	0/19 (0%)
Not circumscribed	Indistinct	9/33 (27%)	4/17 (24%)	4/17 (24%)	2/16 (12.5%)	8/19 (42%)
	Angular/ spiculated	21/33 (64%)	13/17 (76%)	11/17 (64%)	10/16 (62.5%)	9/19 (47%)
	Micro/ macrolobulated	3/33 (9%)	0/17 (0%)	2/17 (12%)	2/16 (12.5%)	2/19 (11%)
Echo pattern						
Complex/hypoechoic		29/33 (88%)	16/17 (94%)	16/17 (94%)	14/16 (87,5%)	19/19 (100%)
Hyper/isoechoic		4/33 (12%)	1/17 (6%)	1/17 (6%)	2/16 (12,5)	0/19 (0%)
Posterior features						
No Posterior Features		13/33 (39%)	5/17 (29%)	3/17 (18%)	3/16 (19%)	2/19 (11%)
Enhancement		0/33 (0%)	1/17 (6%)	3/17 (18%)	6/16 (37.5%)	4/19 (21%)
Shadowing		7/33 (22%)	4/17 (23%)	8/17 (47%)	2/16 (12.5%)	5/19 (26%)
Combined Pattern		13/33 (39%)	7/17 (41%)	3/17 (18%)	5/16 (31%)	8/19 (42%)
Calcifications						
Present		13/33 (39%)	12/17 (71%)	13/17 (76.5%)	12/16 (75%)	4/19 (21%)
Absent		20/33 (61%)	5/17 (29%)	4/17 (23.5%)	4/16 (25%)	15/19 (79%)
Additional features						
Skin changes		4/33 (12%)	6/17 (35%)	6/17 (35%)	2/16 (12.5%)	3/19 (16%)
Edema		24/33 (73%)	10/17 (59%)	12/17 (71%)	9/16 (56%)	14/19 (74%)
Vascularity		28/33 (85%)	17/17 (100%)	16/17 (94%)	15/16 (94%)	18/19 (95%)
Elastography						
Soft		7/33 (21%)	1/17 (6%)	1/17 (6%)	3/16 (19%)	2/19 (10%)
Intermediate		12/33 (36%)	5/17 (29%)	6/17 (35%)	2/16 (12%)	10/19 (53%)
Hard		14/33 (42%)	11/17 (65%)	10/17 (59%)	11/16 (69%)	7/19 (37%)

LA = luminal A; LB HER2− = luminal B without HER2 overexpression; LB HER2+ = luminal B with HER2 overexpression; HER2+ = human epidermal growth factor receptor 2 positive; TNBC = triple negative breast cancer.

**Table 5 jcm-10-05394-t005:** Effect of age on occurrence of specific breast cancer subtypes.

	OR	*p*-Value	CI
LA	1.046	0.005	1.015–1.082
LB HER2−	0.975	0.1744	0.9375–1.0107
LB HER2+	0.987	0.4653	0.9505–1.0228
HER2+	0.988	0.516	0.9508–1.0249
TNBC	0.985	0.3866	0.9501–1.0193

LA = luminal A; LB HER2− = luminal B without HER2 overexpression; LB HER2+ = luminal B with HER2 overexpression; HER2+ = human epidermal growth factor receptor 2 positive; TNBC = triple negative breast cancer. OR = odds ratio, CI = confidence interval.

**Table 6 jcm-10-05394-t006:** Effect of a variable “posterior features” in specific cancer subtypes (HER2+ vs. other cancer subtypes).

HER2+ vs. Other	OR	*p*-Value	CI
Shadowing	0.639	0.6402	0.0786–4.1934
Combined Pattern	1.237	0.7856	0.2747–6.5233
Enhancement	5.750	0.0324	1.2257–32.8005

HER2+ = human epidermal growth factor receptor 2 positive; OR = odds ratio, CI = confidence interval.

**Table 7 jcm-10-05394-t007:** Effect of a variable “posterior features” in aggressive vs. luminal subtypes.

Aggressive vs. Lum	OR	*p*-Value	CI
Shadowing	3.068	0.0539	1.0043–9.9807
Combined Pattern	1.8	0.2776	0.6331–5.3802
Enhancement	29.250	0.0026	4.651–579.583

Lum = luminal; OR = odds ratio, CI = confidence interval; HER2 gene amplification correlates with more common calcifications. The OR for HER2+ and LB HER2+ subtypes are 3.125-fold higher when there are calcifications vs. lack of them (Table 8).

**Table 8 jcm-10-05394-t008:** Effect of calcifications in HER2+ vs. other breast cancer subtypes.

HER2+ (HER2+ and LB HER2+) vs. Other	OR	*p*-Value	CI
Calcifications	3.125	0.03	0.0917–5.87

LB HER2+ = luminal B with HER2 overexpression; HER2+ = human epidermal growth factor receptor 2 positive; OR = odds ratio, CI = confidence interval.

**Table 9 jcm-10-05394-t009:** Effect of lack of calcifications in TNBC breast cancer versus other breast cancer subtypes.

TNBC vs. Other	OR	*p*-Value	CI
Calcifications	0.176	0.0041	0.0469–0.5335

TNBC = triple negative breast cancer; OR = odds ratio; CI = confidence interval.

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
