# Peer review of "Is There a Correlation between Multiparametric Assessment in Ultrasound and Intrinsic Subtype of Breast Cancer?"

_jcm, 2021, doi:10.3390/jcm10225394_

Round 1

Reviewer 1 Report

This manuscript, “Is there a correlation between multiparametric assessment in ultrasound and molecular subtype of breast cancer?” evaluate the relationship of ultrasound features and molecular subtypes of breast cancer (Luminal A, Luminal B HER2-, Luminal B HER2+, HER2+ and TNBC) in 86 patients with 102 breast cancer tumors. Specific features such as incidence of calcification and posterior enhancement were shown to be correlative in particular subtypes and they also identified features that showed no relationship with a subtype of breast cancer (i.e. margin, shape, orientation, elasticity, vascularity). This is a well-described and interesting approach to understanding the relationship of imaging data with underlying tumor biology. However, more details are needed to enable this approach in our studies.

Major

  • It is unclear how the initial cutoffs for some of the features were selected (i.e. Elastography of soft as <80 kPa or enhancement vs non). More methods details are needed or how they indicated to the sonographers to select which features is important.
  • I would like to see the Tsukuba Elasticity score also included. The text indicates it is, but it is unclear how
  • Need to show examples of images for each imaging metric in a nice figure to show the differences between the categories

Minor:

  • Table 1 needs to have some work to keep words on one line (perhaps decrease font size)
  • Table 6 header, has some quotations incorrect
  • Two paragraphs in the discussion have a single sentence. You need at least 3 sentences to make a paragraph

Author Response

Thank you for the Reviewer’s remarks.

It is unclear how the initial cutoffs for some of the features were selected (i.e. Elastography of soft as <80 kPa or enhancement vs non). More methods details are needed or how they indicated to the sonographers to select which features is important.

I would like to see the Tsukuba Elasticity score also included. The text indicates it is, but it is unclear how

Need to show examples of images for each imaging metric in a nice figure to show the differences between the categories

We added the paragraph in Material and Methods about elastograhy and Tsukuba scale. We also gave some examples of hard and soft lesions.

 Minor: Table 1 needs to have some work to keep words on one line (perhaps decrease font size).Table 6 header, has some quotations incorrect. Two paragraphs in the discussion have a single sentence. You need at least 3 sentences to make a paragraph.

We corrected Table 1 and 6. We also corrected discussion.

Reviewer 2 Report

Please define SWE the first time it is used (page 2).

See page 7 and figure 3. The figure does not entirely support the assertion that size correlates with aggressive cancers as the LB HER2- tumors were the exact same size as the "aggressive cancers". There is a trend present, but the text is overstated if LB HER2- is considered non-aggressive.

It is interesting that the posterior enhancement correlates with a 29-fold higher risk of aggressive cancer.  It would have been interesting to at least comment on the posterior enhance characteristics of DCIS or fibroadenoma or apocrine metaplasia.

The title would be more accurate to discuss "intrinsic subtype" rather than "molecular subtype". 

Likewise, it would be a great future project to do this same US analysis with true molecular subtype data from a gene expression profile or with tumor mutation profiling.

Finally, it is worth mentioning that most of the conclusions from this manuscript have been observed by other groups.  To provide novelty to this manuscript, it might be worth mentioning whether any of the patients were screened/imaged by tomosynthesis, MRI, contrast enhanced mammography or other imaging and whether the conclusions from this paper could guide use of advanced imaging modalities (or preclude money wasted on advanced imaging?)

Author Response

Thank you for the Reviewer’s remarks.

Please define SWE the first time it is used (page 2). Corrected.

See page 7 and figure 3. The figure does not entirely support the assertion that size correlates with aggressive cancers as the LB HER2- tumors were the exact same size as the "aggressive cancers". There is a trend present, but the text is overstated if LB HER2- is considered non-aggressive. Although in Figure 3 we present boxplots for each subtype, statistical testing based on logistic regression was performed on the grouping of subtypes (aggressive vs luminal cancers), and the result of that test was significant. Moreover, the Q1 and Q3 values in LB HER2- subtype are noticeably lower than the corresponding values of aggressive cancers, hence our claim. However, because of the aforementioned proximity of median values in LB HER2- subtype and the aggressive subtypes, we softened our assertion in the current version of the manuscript.

It is interesting that the posterior enhancement correlates with a 29-fold higher risk of aggressive cancer.  It would have been interesting to at least comment on the posterior enhance characteristics of DCIS or fibroadenoma or apocrine metaplasia. We added a paragraph in discussion information about posterior enhancement in nonmalignant lesions.

The title would be more accurate to discuss "intrinsic subtype" rather than "molecular subtype". Corrected.

Likewise, it would be a great future project to do this same US analysis with true molecular subtype data from a gene expression profile or with tumor mutation profiling. It is very interesting suggestion. We did not assess gene expression profile or tumor mutation profiling.

Finally, it is worth mentioning that most of the conclusions from this manuscript have been observed by other groups.  To provide novelty to this manuscript, it might be worth mentioning whether any of the patients were screened/imaged by tomosynthesis, MRI, contrast enhanced mammography or other imaging and whether the conclusions from this paper could guide use of advanced imaging modalities (or preclude money wasted on advanced imaging?

In our work we focused on analysis of ultrasound imaging only. In our opinion the advantage of this work was multiparemetric US assessment including sonelastography.

We are not able to correlate our results with other methods (only few patients had MRI examinations, there were no patients imaged with tomosynthesis or contrast enhanced mammography.

Reviewer 3 Report

In this manuscript, Gumowska et. al. have explored the molecular correlation between molecular subtypes and ultrasound in breast cancer. The manuscript is an interesting attempt to explore this area. However, the manuscript has a lot of scope for improvement. Especially, statistical analysis related to logistic regression should be thoroughly expanded. Below are my comments:

  1. In the abstract, authors should use median age, mention the tissue type (FFPE or frozen, etc), expand abbreviations (OR, TNBC, etc.), 95% CI should be mentioned with OR.
  2. All the figures should be improved. The statistical analyses, box and whisker plot should be expanded in the figure legend.
  3. The authors should discuss the logistic regression model. The results of univariate logistic regression should be presented, preferably in a table format.
  4. The coefficients of each variable should be presented in the results and discussed.
  5. Punctuation error in Table 5 title.
  6. Table 1 should be presented in an improved way.

Author Response

Thank you for the Reviewer’s remarks.

In the abstract, authors should use median age, mention the tissue type (FFPE or frozen, etc), expand abbreviations (OR, TNBC, etc.), 95% CI should be mentioned with OR. Corrected.

All the figures should be improved. The statistical analyses, box and whisker plot should be expanded in the figure legend. We corrected all figures and legends were added.  

The authors should discuss the logistic regression model. The results of univariate logistic regression should be presented, preferably in a table format.

The coefficients of each variable should be presented in the results and discussed. We added Table with the coefficients of each variable (supplementary materials).

Punctuation error in Table 5 title. Corrected.

Table 1 should be presented in an improved way. Corrected.

Reviewer 4 Report

Gumowska et al. performed a retrospective study to investigate the correlation of ultrasound features and molecular subtypes of breast cancer in a cohort of 86 patients. With increasing application of molecular profiling to guide therapeutic decision making, it would be beneficial to identify imaging parameters that may associate with / predict the molecular profiles of malignant lesions. The findings of this study are valuable in that regard. The manuscript is well-written, the data have been clearly presented and the conclusions are supported by the data presented in the manuscript.

Specific comments that could be addressed:

Major comments:

  1. There is insufficient data on the clinical parameters of the included patients. Were these tumors primary or recurrent? Did these patients have familial history of breast cancer? How were these patients treated and followed up? Were there any significant differences in terms of these clinical parameters among the included patients?
  2. Details of the immunohistochemistry methodology should be provided as supplementary material.
  3. The authors could present the p-values corresponding to the BIRADS characteristics of specific breast cancer subtypes in Table 3.
  4. The authors could consider adding histology and immunohistochemistry images in a panel as Figures 4 and 5 to support their findings.
  5. Did the authors perform any feature selection? Did any particular group of features associate with particular histological / immunohistochemical subtype?
  6. Why did the authors not include a control group of benign breast lesions in this study?
  7. The authors could present the findings related to association of BIRADS features with histological / IHC subtypes from relevant literature as a table in the manuscript. This will help put the findings of this study in context better.

Minor comments:

  1. Initials of the study radiologist(s) and pathologist(s) who performed the assessments should be provided in the materials and methods section.
  2. The abbreviations ICH and IHC have been used inconsistently. The authors should use IHC as the abbreviation for immunohistochemistry, as this is most frequently used across publications. Abbreviations should be spelled out at the point of introduction in the manuscript and then used consistently throughout the manuscript.
  3. The legend to Figure 3 is missing.
  4. Page 10: TP53 needs to be italicized.
  5. Limitations of the study should not be placed in the conclusion, but in the discussion section.

Author Response

Thank you for the Reviewer’s remarks.

There is insufficient data on the clinical parameters of the included patients. Were these tumors primary or recurrent? Did these patients have familial history of breast cancer? How were these patients treated and followed up? Were there any significant differences in terms of these clinical parameters among the included patients? All patients had primary breast tumors. We focused on US imaging analysis in particular subtypes of breast cancer. We do not have data concerning familiar history of breast cancer. All examined patients were referred to Maria Sklodowska-Curie National Research Institute of Oncology for confirmation of diagnosis and further treatment. We didn’t analyze further patients history.  

Details of the immunohistochemistry methodology should be provided as supplementary material. We added a paragraph in Supplementary materials about the immunohistochemistry methodology.

The authors could present the p-values corresponding to the BIRADS characteristics of specific breast cancer subtypes in Table 3. We added Table with the coefficients of each variable (in Supplementary materials).

The authors could consider adding histology and immunohistochemistry images in a panel as Figures 4 and 5 to support their findings. Corrected

Did the authors perform any feature selection? Did any particular group of features associate with particular histological / immunohistochemical subtype? We tried to select features that could predict a specific cancer subtype – we found a significant differences for calcifications and posterior enhancement. We also tried to combined features and make decision models for specific cancer subtypes (we didn’t find any relationships).

Why did the authors not include a control group of benign breast lesions in this study? We analyzed only patients with breast cancer, we didn’t analyze benign lesions.

The authors could present the findings related to association of BIRADS features with histological / IHC subtypes from relevant literature as a table in the manuscript. This will help put the findings of this study in context better. We added Table with literature review.

Minor comments:

  1. Initials of the study radiologist(s) and pathologist(s) who performed the assessments should be provided in the materials and methods section. Corrected
  2. The abbreviations ICH and IHC have been used inconsistently. The authors should use IHC as the abbreviation for immunohistochemistry, as this is most frequently used across publications. Abbreviations should be spelled out at the point of introduction in the manuscript and then used consistently throughout the manuscript. Corrected

  1. The legend to Figure 3 is missing. Corrected

  1. Page 10: TP53 needs to be italicized. Corrected

  1. Limitations of the study should not be placed in the conclusion, but in the discussion section. Corrected

Round 2

Reviewer 3 Report

The Authors have addressed most of the comments.

Reviewer 4 Report

The authors have adequately addressed all of my comments. I have no further comments.